# Structure and lipid dynamics in the maintenance of lipid asymmetry inner membrane complex of *A. baumannii*

Daniel Mann [1,8], Junping Fan[2,9,10], Kamolrat Somboon[3,10], Daniel P. Farrell [4], Andrew Muenks[4], Svetomir B. Tzokov [1], Frank DiMaio[4], Syma Khalid[3], Samuel I. Miller [2,5,6] & Julien R. C. Bergeron [1,7 ✉]

Multi-resistant bacteria are a major threat in modern medicine. The gram-negative cocco-bacillus *Acinetobacter baumannii* currently leads the WHO list of pathogens in critical need for new therapeutic development. The maintenance of lipid asymmetry (MLA) protein complex is one of the core machineries that transport lipids from/to the outer membrane in gram-negative bacteria. It also contributes to broad-range antibiotic resistance in several pathogens, most prominently in *A. baumannii*. Nonetheless, the molecular details of its role in lipid transport has remained largely elusive. Here, we report the cryo-EM maps of the core MLA complex, MlaBDEF, from the pathogen *A. baumannii*, in the apo-, ATP- and ADP-bound states, revealing multiple lipid binding sites in the cytosolic and periplasmic side of the complex. Molecular dynamics simulations suggest their potential trajectory across the membrane. Collectively with the recently-reported structures of the *E. coli* orthologue, this data also allows us to propose a molecular mechanism of lipid transport by the MLA system.

[1] Department of Molecular Biology and Biotechnology, The University of Sheffield, Sheffield, UK. [2] Department of Microbiology, The University of Washington, Seattle, USA. [3] Department of Chemistry, University of Southampton, Southampton, UK. [4] Department of Biochemistry, The University of Washington, Seattle, USA. [5] Department of Genetics, The University of Washington, Seattle, USA. [6] Department of Medicine, The University of Washington, Seattle, USA. [7] Randall Division of Cell and Molecular Biophysics, King's College London, London, UK. [8]Present address: Ernst-Ruska-Centre 3, Forschungszentrum Jülich, Germany. [9]Present address: Department of Chemical Biology, Peking University, Beijing, China. [10]These authors have contributed equally: Junping Fan, Kamolrat Somboon. ✉email: julien.bergeron@kcl.ac.uk

Gram-negative bacteria are enveloped by two lipid bilayers, separated by the periplasmic space containing the peptidoglycan cell wall. This two-membrane system shields them effectively from a range of antibiotics like Penicillin, and also from chemicals like detergents or enzymes like lysozyme. Since many important bacterial pathogens like *Pseudomonas aeruginosa*, *Campylobacter* or *Acinetobacter* belong to this group of bacteria, they form an important target in modern medicinal research[1]. The two membranes in Gram-negative bacteria have distinct lipid compositions: the inner membrane consists of glycerophospholipids, with both leaflets having similar compositions, while the outer membrane is asymmetric, with an outer leaflet of lipopolysaccharides and an inner leaflet of glycerophospholipids[2] (Fig. 1A). This lipid gradient, depicting the first and most important permeation barrier, is maintained by several machineries, including YebT, PqiB and the multi-component maintenance of lipid asymmetry (MLA) system[3,4], which consists of MlaA present in the outer membrane, the shuttle MlaC in the periplasmic space and the MlaBDEF ABC transporter system in the inner membrane (Fig. 1A). The structure of some of these components have previously been solved: the outer membrane protein MlaA, which was found to form a stable complex with outer membrane porins OmpF and OmpC[5,6], and the periplasmic protein MlaC revealing a hydrophobic pocket for direct lipid transport through the periplasm[3,7]. Low-resolution cryo-EM maps of the MlaBDEF core complex, from *Escherichia coli* (MlaBDEF$_{ec}$)[3] and *Acinetobacter baumannii* (MlaBDEF$_{ab}$)[8] have also been reported, and revealed the overall architecture of the complex, but did not allow to elucidate the molecular details of lipid binding and transport. Opinions about the directionality of lipid transport by the MLA system have been highly controversial, with initial reports suggesting that it recycles lipids from the outer membrane to the inner membrane[3,9,10], but recent results[8,11,12] indicated that it might export glycerophospholipids to the outer membrane. Molecular dynamics simulations of the outer membrane MlaA, and periplasmic MlaC proteins have previously been reported in the literature[6,7,11], but none of the MlaBDEF inner membrane complex.

In this study, we report the structure of the MlaBDEF$_{ab}$ complex in detergent, in three nucleotide states, by single-particle cryo-EM. We also performed molecular dynamics simulations to gain insights into the dynamics of lipids within their observed binding sites. Collectively, this provides important insights into the mechanism of lipid transport by the MLA system, and about the characterization of membrane proteins in detergent.

## Results

**Structure of MlaBDEF$_{ab}$.** We had previously reported the purification of MlaBDEF$_{ab}$ in the presence of the detergent n-dodecyl β-D-maltoside (DDM), and its structure to ~ 8 Å, by single-particle cryo-EM, from data collected on a side-entry 200 kV microscope[8]. In order to improve the resolution of this structure, we collected a dataset of the same complex, in the presence of the non-hydrolizable ATP analogue App-NHp, using a state-of-the-art Titan Krios instrument. Using this better and larger dataset, we were able to refine the structure to ~3.9 Å resolution (Fig. 1B, Table 1, Supplementary Fig. 1 and Supplementary Data 1). This map allowed us to build a de novo atomic model using Rosetta[13] (see 'Methods' for details).

As shown in Fig. 1B, C, the transmembrane multiprotein complex features a 6-fold symmetric assembly of MlaD, with the C-terminal helix forming a basket in the periplasmic space (Fig. 1D), and the N-terminal helix spanning the inner membrane (Fig. 1C). The N-terminal TM helices of MlaD are wrapped around the two MlaE molecules in the membrane, with three MlaD helices interacting asymmetrically with one MlaE, as reported previously[8]. Intriguingly, while MlaE was predicted to have 6 TM helices, we

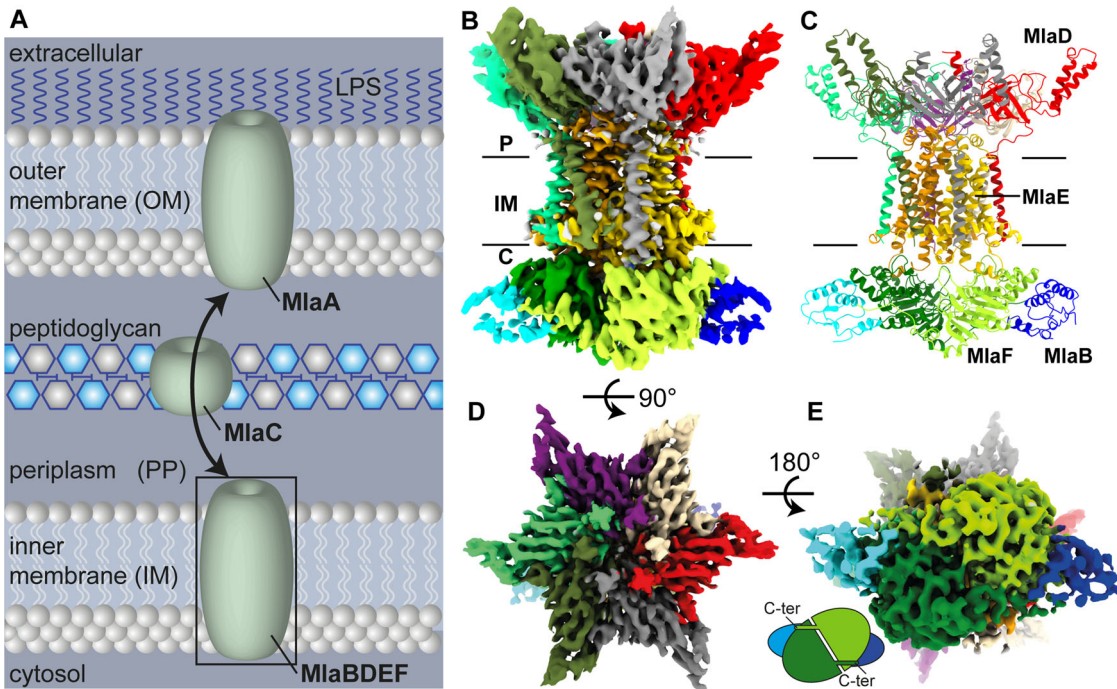

**Fig. 1 Structure of MlaBDEF$_{ab}$. A** Schematic representation of lipid transport by the MlaABCDEF system in Gram-negative bacteria (LPS = Lipopolysaccharide). **B** 3.9 Å Cryo-EM map of MlaBDEF$_{ab}$-AppNHp in β-DDM. **C** Cartoon representation of the MlaBDEF$_{ab}$ atomic model. The complex has a global C2 symmetry, with six copies of MlaD that span the inner membrane from the periplasmic space to the cytosol, two copies of MlaE embedded in the membrane (yellow/gold), two copies of the ATPase MlaF in the cytosol (green), each bound to a copy of MlaB (blue/cyan). **D** Top view of the MlaBDEF$_{ab}$ complex reveals C6 symmetry of MlaD. **E** C-terminal regions of MlaF bind the opposing MlaB subunit via a handshake mechanism.

**Table 1 Cryo-EM data processing and refinement statistics.**

| | AppNHp (EMDB-11082) (PDB: 6Z5U) | Apo (EMDB-11083) | ADP (EMDB-11084) |
|---|---|---|---|
| Data collection and processing | | | |
| Microscope | Titan Krios | Titan Krios | Titan Krios |
| Voltage (kV) | 300 | 300 | 300 |
| Camera | K2 summit | K3 bioquantum | K2 summit |
| Pixel size (Å) | 1.07 | 0.41 | 1.07 |
| Defocus range (μm) | −1 to −2.5 | −0.8 to −2.0 | −1 to −2.5 |
| Total dose (e Å$^{-2}$) | 47 | 40 | 40 |
| Number of micrographs | 2557 | 4737 | 1901 |
| Total particles used | 93,295 | 28,699 | 62,512 |
| Map resolution (Å) | 3.92 | 4.24 | 4.43 |
| *Refinement* | | | |
| Model composition | | | |
| Non-hydrogen atoms | 17,892 | | |
| Protein residues | 2334 | | |
| Ligand atoms | 4 | | |
| B factors (Å$^{-1}$) | | | |
| Protein | 145.24 | | |
| Ligand | 170.31 | | |
| RMS deviations | | | |
| Bond lengths (Å) | 0.008 | | |
| Bond angles (°) | 1.384 | | |
| Validation | | | |
| MolProbity score | 3.43 | | |
| Clashscore | 35.15 | | |
| Poor rotamers (%) | 10.94 | | |
| Ramachandran plot | | | |
| Favoured (%) | 87.45 | | |
| Allowed (%) | 10.94 | | |
| Disallowed (%) | 0.43 | | |

**Table 2 A summary of the MD simulations presented in this paper.**

| System | Temperature (K) | Simulation length (ns) | Membrane composition (number of molecules) | | |
|---|---|---|---|---|---|
| | | | POPE | POPG | CDL |
| Mla_310 | 310 | 500 | 1005 | 268 | 67 |
| Mla_323 | 323 | 500 | 1004 | 267 | 67 |
| Mla_7PE | 310 | 250 | 1012 | 268 | 67 |
| Mla_4PE3PG | 310 | 250 | 1009 | 271 | 67 |
| Mla_ATP | 310 | 250 | 1005 | 268 | 67 |

observe that TM1 does not traverse the membrane, but is monotropically embedded in the inner leaflet, a feature similar to the G5G8 human sterol exporter[14], suggesting similar mechanisms between these complexes, and further supporting the MLA system as a lipid exporter. On the cytosolic side, MlaE is anchored into the ATPase MlaF via the coupling helix situated between TM3 and TM4 (Fig. 1), again similar to the G5G8 sterol exporter complex. MlaF is bound to MlaB away from the nucleotide binding site, similar to the recently reported *E. coli* MlaBF structure[15], with the C-termini of MlaF binding the opposing MlaB subunit by a handshake mode (Fig. 1E). We note, however, that particle classification demonstrated that only ~50% of the particles included MlaB bound to both MlaF, leaving the other 50% bound to only one copy of MlaB (Supplementary Fig. 2). Dual binding of MlaB did not

introduce major structural alterations at the detected resolution, and this observation may correspond to a regulatory role for MlaB or could be due to complex disassembly during sample preparation.

**Lipids spontaneously bind into pockets of cytoplasmic MlaB-DEF$_{ab}$.** In the cryo-EM-derived map of MlaBDEF$_{ab}$, we observed well-defined density in the pocket formed between the MlaE TM1 and two MlaD helices, within the inner leaflet of the lipid bilayer (Fig. 2A, C), which could not be interpreted by protein atoms. We also observed that this pocket is coated by cationic residues, mainly Arg14, Arg47 and Arg234 of MlaE, forming a charged pocket that might attract a lipid head group (Fig. 2D). This observation prompted us to propose that this density corresponds to detergent molecules bound to the complex. As the sample was solubilized in DDM, this density is most likely occupied by the maltoside ring, with the flexible hydrophobic chains extending up into an apolar region of MlaE. In order to confirm if the observed density was consistent with lipid positioning, we performed two independent molecular dynamics simulations (Table 2) with unoccupied binding pockets as starting structures and observed rapid incorporation of bulk lipids during equilibration in both simulations (Fig. 2B, E, F). As an additional test, a further two simulations (Table 2) were performed by removing the lipids that had entered the binding sites, these were also run for 500 ns. Once again lipids rapidly moved to occupy the binding sites. Newly bound lipids remained stably bound thereafter during production runs (Fig. 2F) in all four simulations during 500 ns trajectories. The translational motion of these bound lipids was less than that of the 'bulk' lipids of the model inner membrane (Fig. 2F). In each of the four simulations, two lipids were observed entering the protein, one in each binding site. Interestingly we always observed one PE in one site, and PG in the other. Overall, each lipid within the binding site formed at least two hydrogen bonds with the protein over the course of the simulations. These hydrogen bonds were most often between the phosphate moiety of the lipid head group and arginine residues; R14, R108, R234, R365 and R491 (Supplementary Fig. 7). Other residues which participated in interactions with the lipids were W238 and W491, largely also through headgroup moieties. We propose here that lipids binding in these regions depicts a reasonable first step in lipid export.

**Lipids bind into the periplasmic basket and partially flip.** The periplasmic region of MlaBDEF$_{ab}$ consists mainly of hexameric MlaD forming a basket shape (Fig. 1). Similar to other MCE domain proteins[16], MlaD consists of a central beta sheet motif with a central pore loop that is formed by hydrophobic Leu153/Leu154 in the centre of the C6 symmetric complex (Fig. 3A). We observed that in our map, unattributed density was present between the central pore loops, as well as in the central pore (Fig. 3A). Importantly, this density is not an artefact of the 6-fold symmetry, as it is also resolved when no symmetry was applied during the reconstruction (Supplementary Fig. 8). The presence of lipid molecules in very similar positions were previously reported in the MCE protein YebT[17], which prompted us to postulate that this unattributed density on the MlaD periplasmic region also corresponds to detergent molecules. The basket region has previously been proposed to form the binding site for the periplasmic carrier protein MlaC[5], suggesting that lipids are extracted from this position upon MlaC binding, consistent with the interpretation that these regions of the map correspond to lipid molecules.

Because of this likely important role of lipid dynamics for this region, we next performed 3D variability analysis[18], to identify

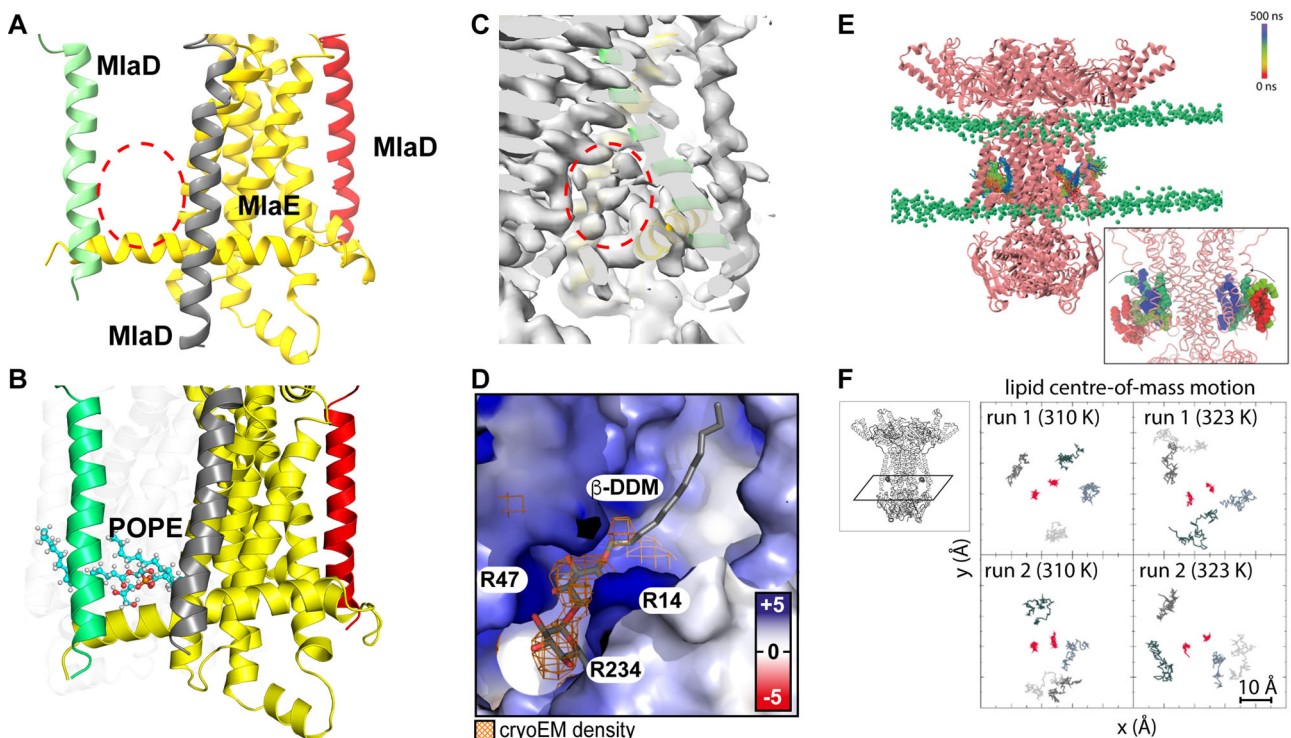

**Fig. 2 Detergent position and dynamics at the MlaD TM–MlaE interface. A–C** Transmembrane region of MlaBDEF$_{ab}$ with one MlaE and three MlaD proteins shown. Red circles indicate density unoccupied by protein residues. Note that at this cut-off the detergent micelle is not visible, but the density attributed to the allocrit is clearly visible. **D** Electrostatic surface potential of modelled inner membrane MlaBDEF$_{ab}$ (blue = +5 kT/e, white = 0 kT/e, red = −5 kT/e) shows a cationic binding site for anionic lipid head groups like DDM and an apolar pocket for the uncharged lipid tails. DDM was modelled inside the previously empty cryo-EM density (orange). **E, F** Molecular dynamics simulations show spontaneous, stable binding of lipids into the cytosolic binding pockets of MlaBDEF. Lipid-free MlaBDEF$_{ab}$ was embedded into an inner membrane model system of 75% POPE, 20% POPG, 5% cardiolipin (headgroups as green spheres, see 'Methods' section for details). Panel (**E**) shows the trajectory of two POPE lipids during a 500 ns simulation, the colour scheme indicates the movement of the lipids as shown in the legend. In this simulation the lipids moved into this location spontaneously during the equilibration process, as shown in the close-up view in the inset in which the red coloured lipids indicate starting positions before the equilibration process. Panel (**F**) shows the motion of the lipids in the *xy* plane (inset) over 500 ns. The red points represent the centre of mass (in *xy*) of the lipids in the binding site, whereas the different shades of grey represent four 'bulk' membrane lipids from each simulation. Data from both independent simulations (r1 and r2) are shown at both temperatures. The data represents 100 frames taken at 5 ns intervals from each trajectory of 500 ns duration. The lipids are confined within an area of 10 × 10 Å for 500 ns, showing this location is favourable.

molecular motions in the MlaD basket. As shown in Supplementary Movies 1 and 2, this analysis revealed that the 6x MlaD basket can be translated and rotated against the MlaBEF transmembrane part, which may play a role in the lipid transport mechanism. Incidentally, this dynamic property likely also limits the achievable resolution for this region of the map.

To further investigate the dynamic properties of the lipids present in the MlaD basket, we set up two systems, one in which we replaced the DDM molecules with 7x PE lipids and one in which we replaced the DDM molecules with 4x PE and 3x PG lipids, and subjected both systems to molecular dynamics simulations (Table 2). PE and PG were chosen here, given they have both been shown to bind to MlaBDEF (*E. coli*) by mass spectrometry[19] and thin-layer chromatography[10]. As shown in Fig. 3, we observed that the peripheral lipids are quite stable within the basket during the course of the simulation. In contrast, the central lipid undergoes a motion within 150 ns of the trajectory. Remarkably, during this motion, we observed lipid flipping (Fig. 3D), with the head group which was modelled away from the central pore operating at an almost 180°, with the polar group buried within the MlaE transporter. This observation likely indicates that we had initially built the lipid in the wrong orientation, and highlights the importance of properly modelling lipid molecules, especially at the intermediate resolution such as

that of our MlaBDEF$_{ab}$ map. In addition, this demonstrates the presence of a large hydrophobic pocket at the MlaD–MlaE interface, which could correspond to the channel for lipid transport.

**Nucleotide binding occurs at the interface of MlaE, MlaF and MlaD.** As indicated above, our structure of MlaBDEF$_{ab}$ was determined in the presence of the non-hydrolizable ATP analogue AppNHp. Accordingly, we observed clear density of the nucleotide and magnesium, within the MlaF binding pocket (Fig. 4A). The nucleotide links the ATPase subunits with the lipid transport domains at the interface of MlaF, MlaE and the N-terminus of MlaD. The resolution was sufficient to map the phosphate binding regions like Walker-A motif around Lys55 of MlaF, as well as binding of the γ-phosphate by Ser51 and His211 (H-loop). The Mg$^{2+}$ ion is coordinated by the Walker-B motif around Asp177 (Fig. 4A) and adenosine ring binding is achieved through Arg26 in the A-loop. As revealed in the previously reported low-resolution map[8], the position of MlaF suggest that the complex is in the substrate-bound conformation.

In order to identify structural changes in the complex associated with ATP binding and hydrolysis, we next determined the structure of MlaBDEF$_{ab}$ without nucleotide (Supplementary Fig. 3), and with ADP (Supplementary Fig. 4). As shown in

Fig. 4A–C, we obtained both structures, at ~4.2 and ~ 4.4 Å, respectively (Table 1 and Supplementary Data 2 and 3). We note that in spite of the nominal global resolution, the map for the complex bound to ADP possesses features largely similar to that of the complex bound to AppNHp. In contrast, the map of the apo complex is less well-resolved, in particular with most TM helices being mostly featureless. This suggests that nucleotide binding stabilizes the overall architecture of the MlaBDEF$_{ab}$ complex.

Nonetheless, we note that the overlay of the MlaBDEF$_{ab}$ maps in the apo, AppNHp-bound and ADP-bound states shows no major structural changes in any of the complex components, other than the nucleotide binding site (Fig. 4D). This indicates that in spite of the nucleotides being bound to the ATPase domain, the conditions used here are not sufficient to trigger the activation of the channel opening.

## Discussion

While this manuscript was under review, three independent groups released the structure of the *E. coli* MlaBDEF complex (MlaBDEF$_{ec}$), in a range of nucleotide states and with various solubilization approaches[19–21].

Comparison of the MlaBDEF$_{ec}$ structure (PDB-ID 7CH0, 7CGN), determined in lipid nanodiscs, and the MlaBDEF$_{ab}$ structure (this study), determined in DDM, reveals an ~10 Ang movement of the MlaD TM helices, accompanied by an ~30° tilt of the MlaE N-terminal helix (Fig. 5A). This movement narrows the lipid binding pocket (Fig. 5). A structural flexibility like this would allow adjustment of MlaBDEF for various lipids. However, as lateral pressure in the membrane plane of a detergent micelle is not comparable to a lipid nanodisc, differences in sample preparation can also cause movement of this hinge. Furthermore, although the MLA system is conserved in Gram-negative bacteria, the sequence identity between the MlaBDEF$_{ab}$ and MlaBDEF$_{ec}$ proteins is around 30–40% identical (depending on the protein, Supplementary Fig. 9), and it is therefore possible that the differences observed between the two structures correspond to variability between bacterial species.

We note that, in one of the aforementioned studies, two ATP-bound conformational states were observed for MlaBDEF$_{ec}$: a more open conformation (*tall state*) and a tightly bound conformation (*close state*), with a shifted nucleotide position. The nucleotide position in MlaBDEF$_{ab}$ in detergent resembles the *tall state* with an open conformation of MlaF (Fig. 5B, C). Our variant analysis did not reveal these two conformations in MlaBDEF$_{ab}$, which might therefore be due to the different orthologues, or the different method for solubilization.

We also emphasize that the detergent binding pockets observed in our MlaBDEF$_{ab}$ structures are also confirmed in the MlaBDEF$_{ec}$ structures, notably the lipids found between the MlaD TM and the N-terminal helix of MlaE[19], and the detergent present in the central cavity[19–21]. For the latter, different exact detergent positions are observed depending on the state and study, but they are largely consistent with the lipid positions we have obtained in the MD simulations, with the charged group buried at the MlaE–MlaD interface. However, our map is the only one with clear detergent density at the interface between MlaD molecules in the periplasmic site. As mentioned above, this could correspond to species specificity, or could be an artefact of the use of detergent for structure determination.

In the light of the multiple structures now available, the structures of MlaBDEF$_{ab}$ reported here support a mechanism for lipid export as summarized in Fig. 6. We observe lipids binding to the inner lipid binding pocket; this region is likely flexible and can possibly adapt to different lipids. MlaE is in very close proximity

to ATP in the closed conformation, possibly describing an additional regulatory mechanism in *A. baumannii*. Structures of MlaBDEF$_{ec}$ also showed a large-scale structural alteration upon tight ATP binding[19] that might be restrained due to the detergent environment in our experiments. We could furthermore observe a central lipid binding site in the periplasmic basket region of MlaD that partially flips into the central channel of MlaE during the course of our MD simulation. Nonetheless, neither our structures of MlaBDEF$_{ab}$ nor the aforementioned MlaBDEF$_{ec}$ could resolve lipid transition between the cytoplasmic lipid binding pocket and the central channel. Similarly, lipid transport from MlaBDEF to MlaC remains elusive. These open questions are also difficult to address with molecular simulations, and it is useful here to reflect on the wider utilities and limitations of MD for studying a system of the size and complexity of the MlaBDEF complex. The utilities of standard, equilibrium MD are clear; the movement of lipids in and around the protein and protein conformational dynamics (within the timescale limitations of MD) can be probed, as we have done here. Any preference for one lipid type over another for being transported by the Mla system is more difficult to determine using standard MD due to limited sampling. Here, free-energy methods are more useful such as those demonstrated by Corey et al. with coarse-grained (CG) force fields[22]. Coarse-grained models also provide a more efficient route to determine more long-range effects of the protein on the lipid environment such as membrane thinning/widening or lipid sorting. These aspects are not the focus of the current study, but will be probed in future work.

It should be emphasized that the structural studies reported here favour an anteretrograde direction for lipid transport by structure similarity, as supported by our previous biochemical data in *A. baumannii*[21] as well as other studies on *E. coli*[12,19,21]. Nonetheless, other assays point towards a retrograde directionality[20,23], and the current structures do not conclusively preclude this. Further structural analyses, in particular in complex with the periplasmic carrier MlaC, will be required to resolve this controversy definitively.

## Methods

**MlaBDEF$_{ab}$ protein expression and purification**. The expression and purification of MlaBDEF$_{ab}$ has been described previously[8]. Briefly, the plasmid encoding the full operon under the control of a T7 promoter was transformed into *E. coli* BL21 DE3 cells and grown at 37 °C until the cell density reached OD (600 nm) = 1.0. The temperature was reduced to 20 °C before induction with 1 mM isopropyl β-D-thiogalactoside (IPTG) and incubation overnight. Cells were harvested using centrifugation at 5000*g* and resuspended in ice-cold buffer A (20 mM Tris-HCl (pH 8.0), 150 mM NaCl, 5% (v/v) glycerol) before disruption in an ultrasonicator on ice (6 cycles; 60 s run, 30 s cool down). Cell debris were pelleted at 17,000*g* for 10 min, and the membrane fraction was separated by centrifugation at 100,000*g* for 1 h. This pellet was resuspended by gentle stirring in Buffer A supplemented with 1% (w/v) dodecyl-β-d-maltopyranoside (DDM) at 7 °C for 1 h. After another centrifugation step at 100,000*g* for 30 min the supernatant was applied to a 5 ml Ni-NTA superflow column (GE Healthsciences) equilibrated with buffer A supplemented with 20 mM imidazole and 0.025% (w/v) DDM. The column was washed with buffer A supplemented with 20 mM imidazole and 0.025% (w/v) DDM before elution with buffer A supplemented with 300 mM imidazole and 0.025% (w/v) DDM was performed. The elute was concentrated to 5 ml and ran through a 16-600 HiLoad Superdex 200 pg gel filtration column (GE Healthcare), preequilibrated with 20 mM Hepes (pH 7.0), 150 mM NaCl, 0.025% (w/v) DDM. Peak fractions were collected, purest fractions were selected using SDS-PAGE and concentrated to 5 mg/ml.

**Cryo-EM sample preparation, data acquisition and image processing**. In all, 4 µl freshly purified 5 mg/ml MlaBDEF$_{ab}$ in 20 mM Hepes (pH 7.0), 150 mM NaCl, 0.025% (w/v) DDM was applied on freshly glow-discharged 300 mesh Quantifoil R2/2 grids, blotted for 3.5 s in a Leica EM-GP plunge freezer at 80% humidity and 4 °C, before getting plunged into liquid ethane at −170 °C. For investigation of the AppNHp and ADP states, the protein was mixed 1:1 with 20 mM of the corresponding nucleotide in 50 mM Hepes (pH 8.0), 150 mM NaCl, 0.025% (w/v) DDM buffer and incubated for 60 min on ice prior to grid preparation.

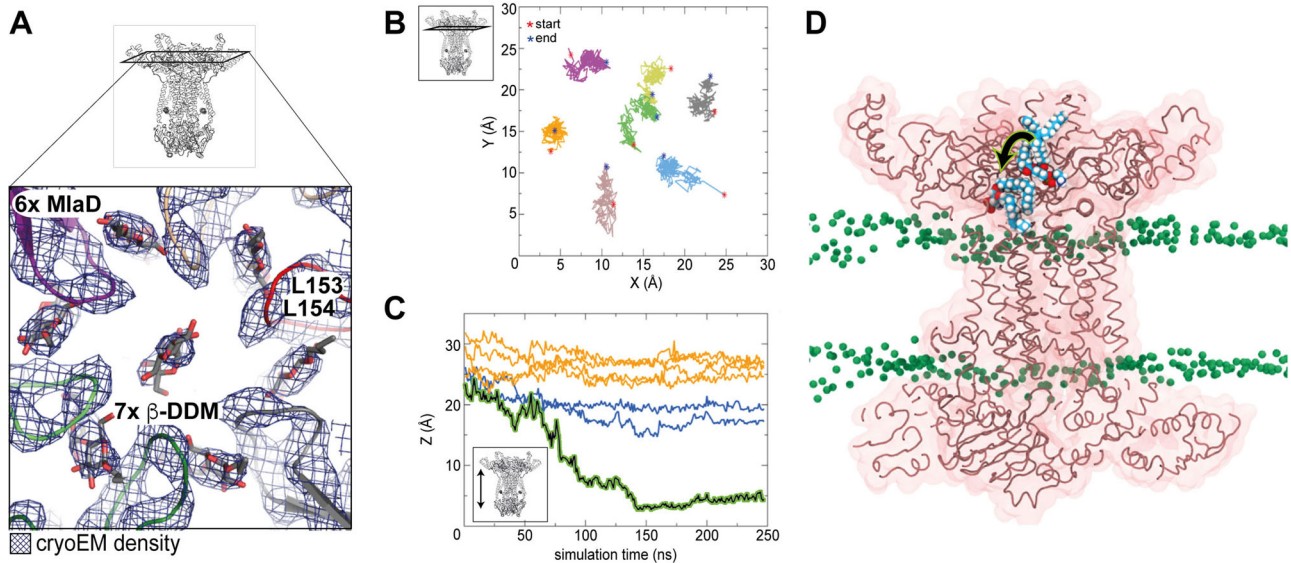

**Fig. 3 Lipid position and dynamics in the MlaD basket. A** Close-up of the periplasmic MlaD basket that houses six peripheral and one central detergent molecule. **B–D** Spontaneous flipping of the centrally bound lipid within the basket region of MlaBDEF$_{ab}$ during 250 ns molecular dynamics simulations. Panel (**B**) shows the centre of mass motion of the seven lipids in the *xy* plane. They are confined to an area of ~5 × 5 Å, indicating this is a high lipid affinity region. Panel (**C**) shows the centre of mass movement of the seven lipids in the *z* dimension as a function of time. The lipid shown in panel (**A**) corresponds to the black curve showing a clear movement towards the cytoplasmic end. Three other lipids (blue) move into this channel but to a lesser extent than the aforementioned lipid, whereas three others (orange) remain close to their starting positions. Simulations from a model at the reported resolution cannot clarify the directionality of the movement, but rather that these regions are conduits for lipids. Panel (**D**) shows a cutaway view of the protein with a POPE lipid at two time points during the simulation, time = 0 ns and 150 ns. The simulation was initiated with seven POPE lipids placed at the periplasmic end of the protein corresponding to the density for detergents in the cryo-EM data (Supplementary Table 2). The lipid which is displaced more towards the cytoplasmic end is from the frame at *t* = 150 ns. The central hydrophobic 'channel' of the protein is a clear conduit for lipids given the spontaneous movement of POPE into this region in just 150 ns.

Micrographs of MlaBDEF$_{ab}$-AppNHp were recorded on a 300 kV Titan Krios microscope with a Gatan K2 Summit detector in counting mode. In all, 2557 movies were recorded with a pixel size of 1.07 Ang in 47 frames with 1 e$^-$/Ang$^2$/frame. In all, 1901 movies of MlaBDEF$_{ab}$-ADP were recorded on the same instrument with a total dose of 40 e$^-$/Ang$^2$. The MlaBDEF$_{ab}$ apo dataset was recorded on a 300 kV Titan Krios equipped with a Gatan K3 bioquantum detector. In all, 4737 micrographs with a pixel size of 0.41 Ang were recorded with a total dose of 40 e$^-$/Ang$^2$ in 50 frames. Data processing was performed in CryoSPARC v2.14.2 (see Supplementary Figs. 1 and 3 for details).

**Model building.** The atomic model of MlaBDEF$_{ab}$ was manually built into the high-resolution regions of the MlaBDEF$_{ab}$-AppNHp map using Coot. Computational intervention was required to build residues 4–38 and 94–134 of MlaD, as well as refine the AppNHp binding site of MlaF. RosettaES[13] was used to model residues 4–38 for each of the 6 MlaD subunits using the manually built model as the starting point. Residues 94–134 were first modelled with Rosetta ab initio[24], which yielded a tightly converged ensemble with a 2-helix topology. The top scoring model from the ab initio predictions was unambiguously docked into the corresponding density for each of the MlaD subunits using UCSF Chimera[25]. Loops were completed and the entire MlaD basket refined using RosettaCM[26] with the context of the cryo-EM density. To refine the AppNHp binding site of MlaF, we first used RosettaCM to hybridize the manually built model with the homologous structures of (3fvq, chain A), and (4ki0, chain B) in the cryo-EM density. Then using homology models and the unexplained density as a guide, a modified version of the Rosetta protocol GALigandDock (unpublished) that uses the cryo-EM data to drive sampling was used to dock the AppNHp molecule. As input to the protocol, a mol2 file of AppNHp was modified using Open Babel (v. 2.4.1)[27] to add hydrogens, charges were assigned with the AM1-BCC charge method in antechamber[28] and a params file was generated with the script main/source/scripts/python/public/generic_potential/mol2genparams.py, which is distributed with Rosetta. Finally, the entire complex was refined using RosettaCM, and magnesium atoms were added by incorporating distance and angle constraints between the Mg atoms and the AppNHp oxygens on the β and γ phosphates during the final minimization. ISOLDE in ChimeraX was used to manually correct for modelling errors. Phenix.real.space.refine was used for refinement and validation. As shown in Supplementary Figs. 5 and 6, the resulting model has an excellent fit to the map density, with clear fit of the side-chains in particular for the TM helices of MlaE and MlaD, which were critical for determining their registry. We note nonetheless that the TM for one of the MlaD molecules is mostly featureless (orange in

Supplementary Fig. 6), in which case we relied on the other two copies to position the helix in the density.

**Molecular dynamics system preparation.** The MlaBDEF$_{ab}$ structure was completed by adding the missing residues using Modeller 9.23 (http://salilab.org/modeller/)[29,30]. The completed protein structure was embedded in an inner *E. coli* inner membrane with dimensions of ~21 × 21 × 18 nm using the CHARMM GUI web server[31–37]. We used a simplified version of the *A. baumannii* inner membrane composition[38]: 75% 1-palmitoyl-2-oleoyl-sn-glycero-3-phosphoethanolamine (POPE), 20% 1-palmitoyl-2-oleoyl-sn-glycero-3-phosphoglycerol (POPG) and 5% 1′,3′-bis[1,2-dioleoyl-sn-glycerol-3-phospho-]-sn-glycerol, cardiolipin (CDL). To relax any steric conflicts within the system generated during set up, energy minimization of 5000 steps was performed on the starting conformation using the steepest descent method. An equilibration procedure followed in which the protein was subjected to position restraints with different force constants. The full equilibration protocol is shown in Supplementary Table 1. The Mla_323 system was set up by removal of lipids that had entered the proposed binding sites during equilibration. An additional set of simulations, Mla_7PE and Mla_4PE3PG were set up in which 7 lipid molecules (PE or PE + PG, details are in Table 1) were positioned such that the lipid head group was overlaid on the maltose moiety of the DDM molecules and the lipid tails were aligned along the DDM hydrocarbon chain using visual molecular dynamics (VMD)[39].

Two independent simulations of each system were performed (denoted r1 and r2 in the main text). The simulation cell had dimensions of 210 × 210 × 180 Å, thus the membrane patch had dimensions of 210 × 210 Å.

**Equilibrium molecular dynamics simulations.** All simulations were carried out with GROMACS 2019.6[40] version (www.gromacs.org) and CHARMM36 forcefield[41]. For the nonbonded interactions and the short-range electrostatics cut-offs of 1.2 nm was applied to the system with the potential shift Verlet cut-off scheme, whereas the long-range electrostatic interactions were treated using the particle mesh Ewald (PME) method[42]. All atoms were constrained using the LINCS algorithm[43] to allow a time step of 2 fs. The desired temperature of either 310 or 323 K was controlled with Nose-Hoover thermostat[44,45] (1.0 ps coupling constant). The pressure was maintained at 1 bar using the Parrinello-Rahman[46] semi-isotropic barostat with a coupling constant of 1.0 ps. Equilibrium MD systems were performed with two repeats per system, where each repeat was run with different initial velocity seeds. The summary of all production runs is shown in

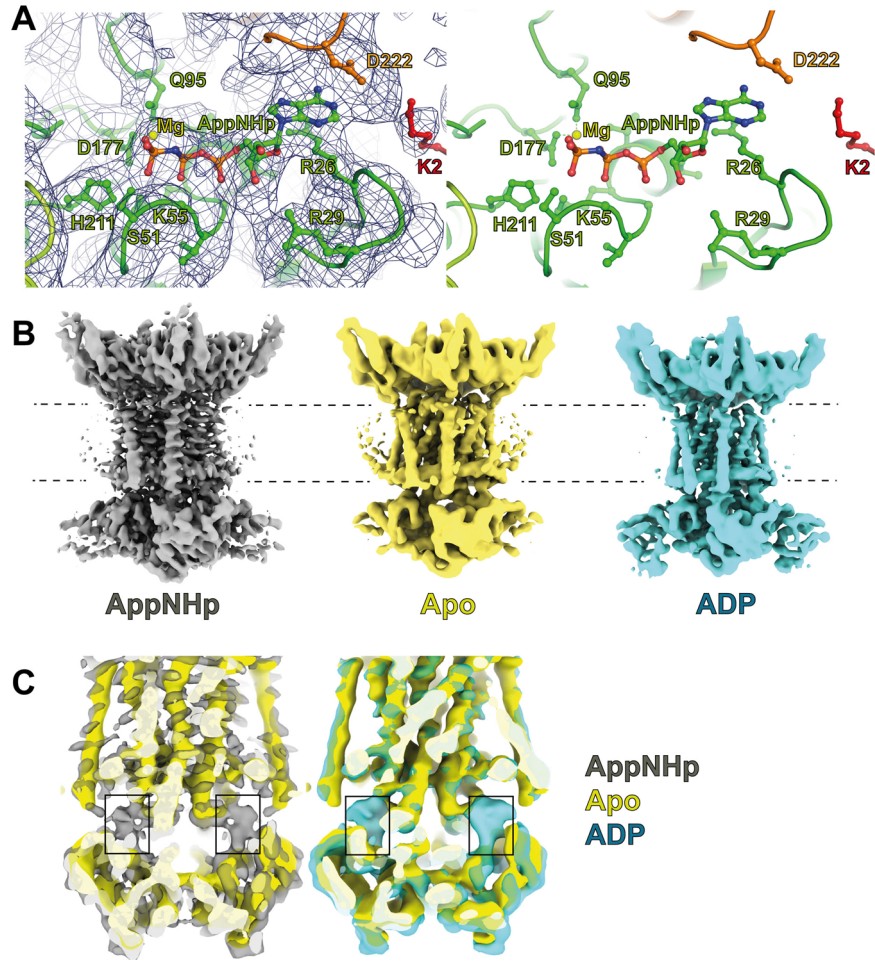

**Fig. 4 Nucleotide position in the MlaBDEF$_{ab}$ structure. A** AppNHp is bound at the interface of MlaE (orange), MlaD (red) and MlaF (green). **B** maps of MlaBDEF$_{ab}$ bound to AppNHp (grey), its *apo* state (yellow) and bound to ADP (cyan). **C** The overlay of the maps unambiguously confirms the presence of nucleotide in the AppNHp and ADP maps, but not in the *apo* map. Nonetheless, no overall structural changes are observed.

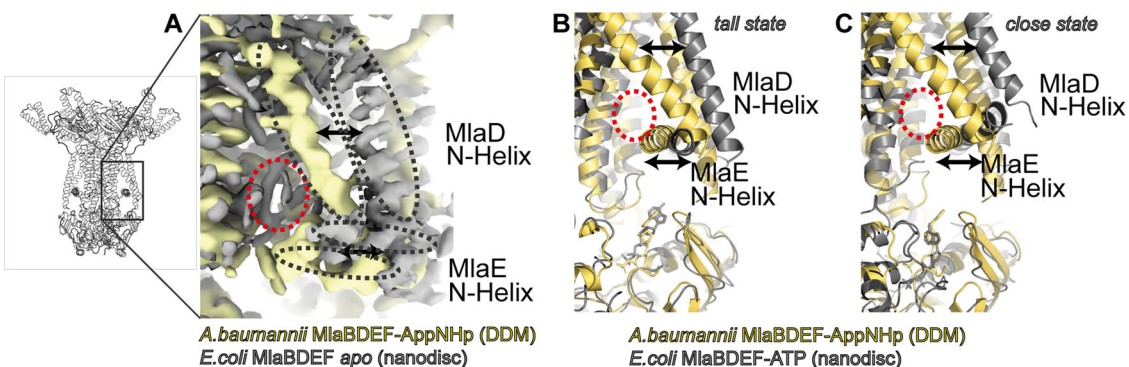

**Fig. 5 Comparison of the MlaBDEF$_{ab}$ and MlaBDEF$_{ec}$ structures.** The cryo-EM map of MlaBDEF$_{ec}$ in lipid nanodiscs[19] and of MLABDED$_{ab}$ in a DDM micelle (this study) show large differences in the lower lipid binding sites. **A** Both the MlaD N-terminal helix and the MlaE N-terminal helix form a much smaller pocket (red circle) in detergent environment (yellow, this study) compared to lipid environment (grey, EMD-30355). Two states were observed in the MlaBDEF$_{ec}$ study; tall (**B**) and close (**C**). Comparison between MlaBDEF$_{ec}$ and MlaBDEF$_{ab}$ shows that the pocket is smaller in both conformations, whereas the nucleotide position is identical in all three structures.

Supplementary Table 2. The results were analysed using GROMACS tools. Molecular graphics were generated using VMD.

**Note added in proof.** Since this paper was submitted to Communications Biology, an independent study reported the structure of the MlaBDEF$_{ab}$ complex, in nanodisc, in the absence of nucleotide[47]. This paper supports the conclusions of this study.

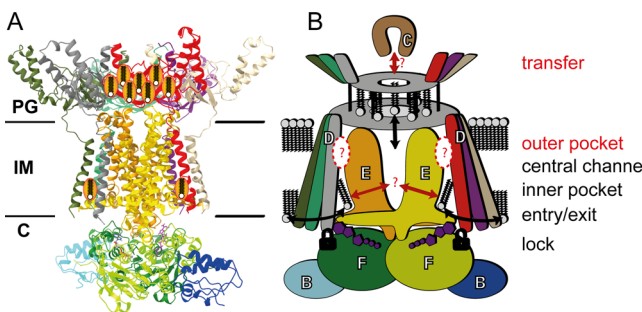

**Fig. 6 Molecular model of lipid transport by the MlaBDEF complex. A**
Structural model, and (**B**) observed degrees of freedom in this study. Lipids
can move freely into the cytosolic paired binding pockets ('entry/exit') that
are very close to the ATP binding sites that directly bind MlaD ('lock').
Lipids can move and even flip inside the basket region ('transfer'). The
elucidated structures could not show how transport from the entry sites to
the basket region is performed and how lipid exchange to/from MlaC is
achieved.

**Reporting summary**. Further information on research design is available in the Nature
Research Reporting Summary linked to this article.

## Data availability

Source data underlying figures are presented in Supplementary Data 1–3. Raw images of
the MlaBDEF$_{ab}$-AppNHp dataset are publicly available under EMPIAR-10425.
Sharpened maps in C2, C1 and C6 symmetry, masks and half-maps of MlaBDEF$_{ab}$-
AppNHp, ADP and apo are publicly available under EMD-11082, EMD-11083 and
EMD-11084, respectively. The MlaBDEF$_{ab}$-AppNHp atomic model is publicly available
under PDB-6Z5U. All other data are available from the corresponding author on
reasonable request.

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

## Acknowledgements
We are thankful for financial support through BBSRC project BB/R019061/1. We acknowledge Diamond Light Source for access and support of the cryo-EM facilities at the UK's national Electron Bio-imaging Centre (eBIC) (under proposal EM-19832). We thank Emma Hasketh and Rebecca Thompson from the Astbury Centre for Structural Molecular Biology Leeds for support during measurements. We acknowledge support from LonCEM, King's College London during measurements. The University of Sheffield FoS cryo-EM facility was used for grid preparation and optimization. We are grateful to Justin Kollman for help with the initial stages of this project. We acknowledge the use of the ARCHER supercomputer via HECBioSim, funded through EPSRC project EP/R029407/1, and the use of the IRIDIS High Performance Computing Facility at the University of Southampton.

## Author contributions
D.M. purified the MlaBDEF complex, performed the cryo-EM experiments, and processed the data, with advice from J.R.C.B.; S.B.T. assisted with cryo-EM grid screening. D.M., D.P. F. and A.M. built the atomic model, with support from J.R.C.B. and F.M.; K.S. and S.K. performed the MD simulation experiments. J.F. made the expression construct and developed the MlaBDEFab purification protocol, and helped with purification, with support from S.I.M. D.M. and J.R.C.B. wrote the manuscript, with comments from all the authors.

## Competing interests
The authors declare no competing interests.
