## [Peer Review File. · Communications Biology]

Reviewers' comments:

Reviewer #1 (Remarks to the Author):

The authors summarized their experimental and computational approach to provide a much more detailed characterization of the MLA complex. Their focus is on the component of the complex in the inner membrane of gram-negative bacteria, MlaBDEF. Their main goal was to provide a better resolution structure to elucidate the interaction and mechanism of lipid transport of this complex. This work is an effort to advance the structural knowledge of the MLA complex for the development of novel therapeutics against gram-negative bacteria.

The manuscript highlights new insights and an impressive $\sim 4\text{\AA}$ improvement in resolution of the structure of the MLA complex using cryo-EM. The authors describe the experimental and computational methods they used to determine the proteins' sequence and spatial conformation. They compared and validated their findings against current and past models for this complex, and provided sound discussion of the similarities and differences with work that was being published during the time of manuscript preparation. Finally, they describe the newest and, to my understanding, the most important contribution their work, which is the location of lipid pockets in the MLA complex and the interaction of lipids with two different regions, with different extent and dynamics of interaction. They extended their experimental observations with long and stable molecular dynamics simulations that provide further details on the lipid-protein dynamics, and predictions as the role of the protein in lipid transport and translocation between the double bilayer bacterial envelope. The combination of experiment and simulation is definitely a strength in the manuscript, and provides robust support for the conclusions discussed by the authors.

Overall the manuscript is well prepared, clear in the objective of the study and contents of the paper. The conclusions definitely contribute to the understanding of lipid transport in/by the MLA complex, and the bigger picture of transmembrane protein dynamics. Some minor items I would like the authors to address are:

1. Expand some of the motivation of the work in the introduction, rather than diving directly into the known and unknowns of the system at hand; though this may be due to space constraints.
2. Proof read the manuscript, as there are a few minor typos such as words typed together.
3. The figure caption of Figure 6 is confusing as it seems to indicate there should be a "C" panel, please fix.
4. Provide some context in terms of the modeling and simulation efforts to study this complex; are there previous simulation studies of this complex, or is this work among the first to examine this particular transporter? If so, also highlight that fact.
5. Provide a supplementary table in which the components of the simulation system are listed; membrane composition is provided in the main text, but it would be useful to have information such as simulation box size (and membrane patch, x/y dimensions), and water / ion molecules used.
6. What is the major challenge or limitation of the simulation studies? Along the lines of system size, energy barriers to explore lipid-protein interactions or protein conformational changes, membrane model composition, i.e. lipid species vs. the detergent molecules used in the experimental setting, etc. The manuscript states the MLA complex may have a lipid binding pocket that accommodates different lipid species, is this channel known to transport or flip a specific set of lipids between the IM and OM of gram-negative bacteria? For example, charged or neutral; and why was POPE selected as a test case to examine the likelihood of a lipid to bind the central cavity in the periplasmic side of the protein?

Reviewer #2 (Remarks to the Author):

Mann et al. present a really nice set of EM maps and a PDB structure of MlaBDEF from the Gram-negative pathogen *Acinetobacter baumannii*. This complex is part of a fascinating system that enables the periplasmic transport of lipids across the periplasmic space. This is a hot and competitive field with three other groups releasing structures of this complex in eLife, NSMB and Cell Discovery. Many of these papers detail maps/structures that are higher resolution than those reported here.

The paper is well written, with clear outcomes. There are aspects that I detail as part of this review that I ultimately think will strengthen the paper and allow better evaluation of the work by the reader.

These aspects predominantly rest with the figures, where I provide a few suggested changes:
Figures:

Figure 1 - 1A is a bit roughly drawn. Perhaps this NPG journal could use their usual cell envelope styling to improve this image. The dimensions of MlaC need to be reduced. Figs 1B-E are nicely presented.

Figure 2 - This figure needs work. I am not convinced by what is shown in Fig 2B-D. It could easily be noise and at the same time, it is not surprising that there might be detergent density around the structure. What is more meaningful biologically is that the MD simulations reveal that this site is a lipid binding site. I would encourage the authors to condense the data shown in 2A-D or include in SI, and pull more out of the simulation data. What are the residues coordinating the bound lipid? Is this selective for PE vs PG vs cardiolipin - here both sites are occupied by PE. I would also suggest that the phosphate headgroup should be better coloured red or orange to reflect the conventional CPK colour scheme for oxygen/phosphorus. There is no description in the legend at present as to what the green spheres show. The insert would be better as a panel in its own right. How do the confined dynamics in 2F compare with the motions of the other lipids in the membrane (or standard lipid diffusion?). Given x and y are the same units of measurement, this figure should be square. How many repeats are performed for the simulations?

Figure 3 - 3A is nicely presented. I think an image of the starting set-up of MlaBDEF with lipids in place of the detergents would be useful for the reader. 3B, like 2F, needs to be square. Ultimately, if the colour of the plot could match the image of the coordinates that would help the reader more quickly interpret what is presented.

My initial interpretation of the black trace of Fig 3C was that full membrane translocation was occurring. Why not colour the traces the same as Fig3B?

It is then confusing that the lipid that is the focus of Fig 3D is moving in the opposite z-direction. I have difficulty with the slice in evaluating where this lipid is at the start? Is it the central lipid or one of the six around the sides? How does this compare with the lipids resolved in the other published MLA structures?

Figure 4 - A looks nice. It would be great if the CPK colour scheme could be used for the coordinating residues, so one can quickly pick out the types of residues coordinating the AppNhp. It would be fantastic to model the AppNhp as ATP and perform an MD simulation of this with Mg to show the interactions made between the coordinates; and that this is faithful to the Cryo-EM structure.

Figure 5 is a nice comparison between the Ab structure and that from Ec. This figure could be improved by using the bound lipid(s) from the simulations to show how the site changes in the two structures. Could the nanodisc be inducing the conformation observed around the E. coli structure? It is tightly wrapped around the structure.

Figure 6 - I think this would be best displayed with bound atomic lipids from the MD simulations rather than the clipart graphic. This would therefore provide a better differentiation between this paper's data and the schematic shown in B.

The image resolution of the protein images in Fig S3 and S4 is rather low.

There are changes in style of the figure legends from eg (A), Panel A or A: This should be standardised in the revision.

Other comments:

The statement on line 26: "It also contributes to broad-range antibiotic resistance in several pathogens, most prominently in *A. baumannii*." needs a reference/evidence.

Although there are three density maps deposited there is only one PDB structure. I think this should be explicitly stated in the abstract, as my initial interpretation was that there would be 3 PDB entries.

On line 191 "It should be emphasized that the structural studies reported here favor an anteretrograde direction for lipid transport" – could you elaborate on this? What specifically about the structure suggests this? The schematic Fig 6B should clearly convey this, if the data suggests anteretrograde transport.

The paper would also benefit from more functional dissection of the structures and assessment of the structures and simulations in light of previous publications. At present the discussion is very much structure focussed.

Minor Comments:

- In the author list there is a erroneous space between Khalid and Frank.
- Dimaio is usually written DiMaio in his other publications, not sure if that needs resolving.
- Gram should be capitalised as it's named after Hans Christian Gram.

Reviewer #3 (Remarks to the Author):

The article by Mann et al., describes the structure of the A. baumannii maintenance of lipid asymmetry inner membrane complex (MlaBDEF).

This structure is a major improvement from the first cryoEM reconstruction published in 2019 (reference 7). The authors are now able to build missing regions and clarify secondary structure elements notably of MlaE helices, with clear side chains in some of them. The structural organization is reminiscent of sterol transporter therefore providing another clue to the possible lipid transporter function of MLA. Detergent molecules were identified in 2 locations in the inner membrane and in the periplasmic portion of MlaD. Molecular dynamics simulations suggest that lipids could stably occupy these sites.

3 structures of the complex were determined using a non-hydrolysable analog of ATP, ADP or no nucleotide. Despite minor differences, the 3 complexes are globally superimposable and resemble the "tall state" observed in a recently solved E. coli homologous complex. Molecular motions of the MalD basket/crown are observed relative to the inner membrane domain. Taking into account E. coli homologous structures in different conformations, they propose a putative mechanism for lipid transport.

In the abstract, the authors describe the rationale of studying A. baumannii as a major pathogen regarding resistance to antibiotics. The article would be strengthened if more information was provided on the specific sequence/structure relationship compared with other species, and in particular E. coli.

The cryoEM data processing is detailed and conducted properly. This article is well written and this study should be of interest in the field and accessible to a larger readership. Several questions should be addressed.

Specific questions or suggestions:

- Page 5 line 122: the authors state that densities of lipid/detergent molecules are visible in the cryoEM maps where no symmetry was used in the reconstruction. It would be useful if the the authors provide a supplemental figure showing both reconstructions showing this region.

- Page 8 line 203: the authors mention that "our map is the only one with clear lipid density at the interface between malD molecules". According to figure 3 and the explanations page 5, those are DDM molecules. Throughout page 5 "lipid" appear instead of "DDM", this is confusing regarding what is actually seen in the experimental data.

- In the previous article (reference 7) the authors had identified glycerophospholipids lipids bound to the purified complex MlaC-D. In the present study, the authors have identified the binding of DDM molecules in the inner membrane region and in the MlaD basket reminiscent of lipid binding sites. Is the position of lipids in the MlaD basket compatible with an interaction with MlaC, or are they evicted upon MalC binding?

- the authors comment on the sequence identity between E.coli and A. baumannii that could explain the structural difference on the MalE helix of the lower lipid binding sites (Figure 5). However they propose a common mechanism where this cavity plays a common role in lipid transport (Figure 6). How different is the sequence conservation in this area and how much the authors would expect this would impact the mechanism ?

- the authors introduce the study by mentioning that the MLA contributes to antibiotic resistance most prominently in A. baumannii. Is there any structural/sequence feature that would explain to some extent this specificity ? It would be interesting to show targeted sequence alignments related to structural elements which might highlight differences between E.coli and A. baumannii.

- related to the previous 2 questions, Page 6 line 133-134 : The authors explain that the MalD crown undergoes some conformational fluctuation as shown by the processing using 3D variability analysis in CryoSPARC (supplemental figure 1) and illustrated in the movies. The authors mention that this flexibility explains the lower resolution in this region. The study by Chi et al , Cell Res. 2020 (reference 20) used adaptive masks with C6 symmetry in Relion to get better resolution on this region. It may be interesting to improve the resolution of this domain to examine the role of species-specific positions, if a significant difference in primary sequence with E.coli is observed in this region.

Other Remarks:

- Figure 1: from the figure on panel B it is not easy to identify MlaE. Maybe a coloring of the MlaE name with the same color as the 2 subunits would help better understand the architecture of the complex. Same comment for MlaF and MlaB. The orientation are well chosen and this figure will be otherwise informative once the subunits are properly identified

- Figure 4A: the Mg ion could be indicated and be displayed in another colour

- Figure 6: panel C is missing and the legend mentions the "transport" step whereas panel B mentions "transfer".

- Supplemental figure 3 and 4: panels A, B, C are not specified in the legend. In addition the particles are difficult to distinguish in image A. A selection of 2D classes would be more informative.

- The authors refer alternatively to the 6-mer head of MalD as a crown, or basket, it would more consistent to stick to one name.

- page 7 line 176 (discussion): the authors mention "while this manuscript was in review", maybe it would be more relevant to say "while this manuscript was in preparation" in the context of this submission ?

Reviewer #1 (Remarks to the Author):

The authors summarized their experimental and computational approach to provide a much more detailed characterization of the MLA complex. Their focus is on the component of the complex in the inner membrane of gram-negative bacteria, MlaBDEF. Their main goal was to provide a better resolution structure to elucidate the interaction and mechanism of lipid transport of this complex. This work is an effort to advance the structural knowledge of the MLA complex for the development of novel therapeutics against gram-negative bacteria.

The manuscript highlights new insights and an impressive $\sim 4\text{\AA}$ improvement in resolution of the structure of the MLA complex using cryo-EM. The authors describe the experimental and computational methods they used to determine the proteins' sequence and spatial conformation. They compared and validated their findings against current and past models for this complex, and provided sound discussion of the similarities and differences with work that was being published during the time of manuscript preparation. Finally, they describe the newest and, to my understanding, the most important contribution their work, which is the location of lipid pockets in the MLA complex and the interaction of lipids with two different regions, with different extent and dynamics of interaction. They extended their experimental observations with long and stable molecular dynamics simulations that provide further details on the lipid-protein dynamics, and predictions as the role of the protein in lipid transport and translocation between the double bilayer bacterial envelope. The combination of experiment and simulation is definitely a strength in the manuscript, and provides robust support for the conclusions discussed by the authors.

Overall the manuscript is well prepared, clear in the objective of the study and contents of the paper. The conclusions definitely contribute to the understanding of lipid transport in/by the MLA complex, and the bigger picture of transmembrane protein dynamics. Some minor items I would like the authors to address are:

1. Expand some of the motivation of the work in the introduction, rather than diving directly into the known and unknowns of the system at hand; though this may be due to space constraints.

We are grateful for this reviewer's supporting comments. We have expanded motivation for work on Gram-negative bacteria (lines 37-41) and the MLA system (lines 60-63) in the introduction.

2. Proof read the manuscript, as there are a few minor typos such as words typed together.

We apologize for these, and have hopefully corrected most of them.

3. The figure caption of Figure 6 is confusing as it seems to indicate there should be a "C" panel, please fix.

We have fixed the figure caption.

4. Provide some context in terms of the modeling and simulation efforts to study this complex; are there previous simulation studies of this complex, or is this work among the first to examine this particular transporter? If so, also highlight that fact.

We have now provided contextual information with references within the introduction (lines 60-63). To our knowledge, this manuscript contains the first molecular dynamics study of MlaBDEF.

5. Provide a supplementary table in which the components of the simulation system are listed; membrane composition is provided in the main text, but it would be useful to have information such as simulation box size (and membrane patch, x/y dimensions), and water / ion molecules used.

We have now included a table with this information (Table 2) in the methods section of the manuscript.

6. What is the major challenge or limitation of the simulation studies? Along the lines of system size, energy barriers to explore lipid-protein interactions or protein conformational changes, membrane model composition, i.e. lipid species vs. the detergent molecules used in the experimental setting, etc. The manuscript states the MLA complex may have a lipid binding pocket that accommodates different lipid species, is this channel known to transport or flip a specific set of lipids between the IM and OM of gram-negative bacteria? For example, charged or neutral; and why was POPE selected as a test case to examine the likelihood of a lipid to bind the central cavity in the periplasmic side of the protein?

We have now provided a perspective on major limitations of the MD simulations within the discussion section (Lines 242-254). PE was chosen as it is a major component of the *A. baumannii* inner membrane. We had previously shown that the MLA complex carries both PE and PG (Kamischke et al., eLife), which has since been confirmed by other studies.

We have now performed additional simulations in which a combination of PE and PG lipids are placed in the central cavity, to test the behaviour of Zwitterionic (PE) and anionic (PG) headgroups.

Reviewer #2 (Remarks to the Author):

Mann et al. present a really nice set of EM maps and a PDB structure of MlaBDEF from the Gram-negative pathogen *Acinetobacter baumannii*. This complex is part of a fascinating system that enables the periplasmic transport of lipids across the periplasmic space. This is a hot and competitive field with three other groups releasing structures of this complex in eLife, NSMB and Cell Discovery. Many of these papers detail maps/structures that are higher resolution than those reported here.

The paper is well written, with clear outcomes. There are aspects that I detail as part of this review that I ultimately think will strengthen the paper and allow better evaluation of the work by the reader.

These aspects predominantly rest with the figures, where I provide a few suggested changes:

Figures:

1. Figure 1 - 1A is a bit roughly drawn. Perhaps this NPG journal could use their usual cell envelope styling to improve this image. The dimensions of MlaC need to be reduced. Figs 1B-E are nicely presented.

We acknowledge this reviewer for his/her constructive comments and suggestions. We have modified Figure 1 A and hope the updated version appears more polished.

2. Figure 2 – This figure needs work. I am not convinced by what is shown in Fig 2B-D. It could easily be noise and at the same time, it is not surprising that there might be detergent density around the structure.

We emphasize that the detergent micelle at the chosen cut-off is not visible in Fig. 2 C, but the density that was interpreted as detergent/lipid is clearly visible. While this is difficult to illustrate within the figure, the PDB, EMDB and EMPIAR data has now been released, and will allow to directly visualize this.

What is more meaningful biologically is that the MD simulations reveal that this site is a lipid binding site. I would encourage the authors to condense the data shown in 2A-D or include in SI, and pull more out of the simulation data. What are the residues coordinating the bound lipid? Is this selective for PE vs PG vs cardiolipin – here both sites are occupied by PE.

We have performed the suggested additional analyses, and have included data on the coordinating residues (largely arginines) in the manuscript. We find both PE and PG in the binding sites, but never cardiolipin. We have added text to clarify the nature of the lipids in the binding sites.

I would also suggest that the phosphate headgroup should be better coloured red or orange to reflect the conventional CPK colour scheme for oxygen/phosphorus. There is no description in the legend at present as to what the green spheres show. The insert would be better as a panel in its own right. How do the confined dynamics in 2F compare with the motions of the other lipids in the membrane (or standard lipid diffusion?). Given x and y are the same units of measurement, this figure should be square.

We have modified the figure to show that the motion of the lipids in the binding site is more confined compared to those further away from the protein ('bulk' lipids). In this instance, it is not relevant to compare diffusion rates given the small number of lipids in the binding site would produce very noisy data. We have instead plotted lateral motion, shown in the revised Figure 2F.

How many repeats are performed for the simulations?

We repeated each simulation twice; this information has now been added to the materials and methods section of the manuscript (lines 356-358). We apologize for this omission.

3. Figure 3 – 3A is nicely presented. I think an image of the starting set-up of MlaBDEF with lipids in place of the detergents would be useful for the reader. 3B, like 2F, needs to be square. Ultimately, if the colour of the plot could match the image of the coordinates that would help the reader more quickly interpret what is presented.

My initial interpretation of the black trace of Fig 3C was that full membrane translocation was occurring. Why not colour the traces the same as Fig3B?

We have modified Fig 3B as suggested.

It is then confusing that the lipid that is the focus of Fig 3D is moving in the opposite z-direction. I have difficulty with the slice in evaluating where this lipid is at the start? Is it the central lipid or one of the six around the sides? How does this compare with the lipids resolved in the other published MLA structures?

We have corrected the z-direction in this figure, and updated the figure caption to indicate that it was indeed the central lipid that was moving.

4. Figure 4 – A looks nice. It would be great if the CPK colour scheme could be used for the coordinating residues, so one can quickly pick out the types of residues coordinating the AppNHp. It would be fantastic to model the AppNHp as ATP and perform an MD simulation of this with Mg to show the interactions made between the coordinates; and that this is faithful to the Cryo-EM structure.

We emphasize that accurate molecular dynamics simulation of a non-covalently bound ATP molecule from a 3.9 Ang structure is challenging, at least. The complex interactions between Mg²⁺, dispersion-driven ring binding and critical water molecules are difficult to model and certainly more complicated compared to hydrophobic lipid-binding as shown in our other simulations. We note that even in MD studies from X-Ray structures of the prototype ABC transporter MsbA the authors had difficulties with very loose ring binding (See Furuta et al., Biochemistry, 2014). Nevertheless, we performed two runs of unrestrained 250 ns molecular dynamics simulations with ATP bound to the active sites of MlaBDEF. While the overall protein structure remains intact and the nucleotide remains inside the active site, we observe that the adenosine ring loses its coordination and flips (see below) during the course of the simulation, while the Mg²⁺-triphosphate remains stably bound. This could be easily overcome with coordination restraint (as often used in molecular dynamics studies of nucleotides), however we argue that this would not add any significant evidence towards supporting that there is no significant changes in nucleotide is positioned between AppNHp-bound, ADP-bound and apo structures in Fig 4.

5. Figure 5 is a nice comparison between the Ab structure and that from Ec. This figure could be improved by using the bound lipid(s) from the simulations to show how the site changes in the two structures. Could the nanodisc be inducing the conformation observed around the E. coli structure? It is tightly wrapped around the structure.

We agree that most likely the difference between lipid nanodisc and detergent is responsible for the observed structural difference. We have indicated this in the results section (lines 207-209)

6. Figure 6 – I think this would be best displayed with bound atomic lipids from the MD simulations rather than the clipart graphic. This would therefore provide a better differentiation between this paper's data and the schematic shown in B.

We agree with this reviewer that using atomic coordinates would be more accurate. However, we note that due to the multiple lipid molecules found in the structure, and their dynamics observed in the MD simulations, it is very challenging to illustrate their position with an atomic model. For this reason, we have chosen to keep schematic representations for the lipids, in figure 6a. We hope this is acceptable to this reviewer.

7. The image resolution of the protein images in Fig S3 and S4 is rather low.

We apologize for this; we have provided figures with improved resolution in the revised manuscript.

8. There are changes in style of the figure legends from eg (A), Panel A or A: This should be standardised in the revision.

We have revised the captions as suggested.

Other comments:

9. The statement on line 26: "It also contributes to broad-range antibiotic resistance in several pathogens, most prominently in *A. baumannii*." needs a reference/evidence.

We cannot include references in the abstract; however, we have added a corresponding reference in the introduction.

10. Although there are three density maps deposited there is only one PDB structure. I think this should be explicitly stated in the abstract, as my initial interpretation was that there would be 3 PDB entries.

We changed "structures" to "maps" in the abstract, to clarify this. While we could deposit three PDB coordinates, we feel that it is not really adding anything, due to the absence of structural changes in all three maps, at the obtained resolution.

11. On line 191 "It should be emphasized that the structural studies reported here favor an anteretrograde direction for lipid transport" – could you elaborate on this? What specifically about the structure suggests this? The schematic Fig 6B should clearly convey this, if the data suggests anteretrograde transport.

We have replaced the highlighted sentence, to "favor an anteretrograde direction for lipid transport by structure similarity", as discussed in the results section.

12. The paper would also benefit from more functional dissection of the structures and assessment of the structures and simulations in light of previous publications. At present the discussion is very much structure focussed.

We included the suggested changes

Minor Comments:

- In the author list there is a erroneous space between Khalid and Frank.
- Dimaio is usually written DiMaio in his other publications, not sure if that needs resolving.
- Gram should be capitalised as it's named after Hans Christian Gram.

We included the suggested changes

Reviewer #3 (Remarks to the Author):

The article by Mann et al., describes the structure of the *A. baumannii* maintenance of lipid asymmetry inner membrane complex (MlaBDEF).

This structure is a major improvement from the first cryoEM reconstruction published in 2019 (reference 7). The authors are now able to build missing regions and clarify secondary structure elements notably of MlaE helices, with clear side chains in some of them. The structural organization is reminiscent of sterol transporter therefore providing another clue to the possible lipid transporter function of MLA. Detergent molecules were identified in 2 locations in the inner membrane and in the periplasmic portion of MlaD. Molecular dynamics simulations suggest that lipids could stably occupy these sites.

3 structures of the complex were determined using a non-hydrolysable analog of ATP, ADP or no nucleotide. Despite minor differences, the 3 complexes are globally superimposable and resemble the “tall state” observed in a recently solved *E. coli* homologous complex. Molecular motions of the MlaD basket/crown are observed relative to the inner membrane domain. Taking into account *E. coli* homologous structures in different conformations, they propose a putative mechanism for lipid transport.

In the abstract, the authors describe the rationale of studying *A. baumannii* as a major pathogen regarding resistance to antibiotics. The article would be strengthened if more information was provided on the specific sequence/structure relationship compared with other species, and in particular *E. coli*.

The cryoEM data processing is detailed and conducted properly. This article is well written and this study should be of interest in the field and accessible to a larger readership. Several questions should be addressed.

Specific questions or suggestions:

- Page 5 line 122: the authors state that densities of lipid/detergent molecules are visible in the cryoEM maps where no symmetry was used in the reconstruction. It would be useful if the authors provide a supplemental figure showing both reconstructions showing this region.

We thank this reviewer for their helpful suggestions. We have now included Supplemental Figure 8 containing three reconstructions with C1, C2 and C6 symmetry applied.

- Page 8 line 203: the authors mention that “our map is the only one with clear lipid density at the interface between mlaD molecules”. According to figure 3 and the explanations page 5, those are DDM molecules. Throughout page 5 “lipid” appear instead of “DDM”, this is confusing regarding what is actually seen in the experimental data.

We agree that this was confusing and changed “lipid” to “detergent” in this paragraph

- In the previous article (reference 7) the authors had identified glycerophospholipids lipids bound to the purified complex MlaC-D. In the present study, the authors have identified the binding of DDM molecules in the inner membrane region and in the MlaD basket reminiscent of lipid binding sites. Is

the position of lipids in the MlaD basket compatible with an interaction with MlaC, or are they evicted upon MlaC binding?

Since MlaC was not present in this study, we do not have any additional evidence on the MlaC-MlaBDEF interaction, especially since the C-terminal ~30 aminoacids of MlaD that are directly facing MlaC could not be resolved in our map. Because of this, is it difficult to speculate on which lipid(s) is taken up by MlaC. Further structural studies, on the complex bound to MlaC, will be required to address this. We have clarified it in the discussion.

- the authors comment on the sequence identity between E.coli and A. baumannii that could explain the structural difference on the MlaE helix of the lower lipid binding sites (Figure 5). However they propose a common mechanism where this cavity plays a common role in lipid transport (Figure 6). How different is the sequence conservation in this area and how much the authors would expect this would impact the mechanism ?

- the authors introduce the study by mentioning that the MLA contributes to antibiotic resistance most prominently in A. baumannii. Is there any structural/sequence feature that would explain to some extent this specificity ? It would be interesting to show targeted sequence alignments related to structural elements which might highlight differences between E.coli and A. baumannii.

We added Supplemental Figure 9 with a pairwise sequence alignment of A.baumannii and E.coli MlaD and MlaE and highlighted the areas that make up the proposed binding cytosolic lipid binding pocket. While the N-terminal regions are less conserved the C-terminal helix of MlaE is almost identical sequence-wise.

- related to the previous 2 questions, Page 6 line 133-134 : The authors explain that the MlaD crown undergoes some conformational fluctuation as shown by the processing using 3D variability analysis in CryoSPARC (supplemental figure 1) and illustrated in the movies. The authors mention that this flexibility explains the lower resolution in this region. The study by Chi et al , Cell Res. 2020 (reference 20) used adaptive masks with C6 symmetry in Relion to get better resolution on this region. It may be interesting to improve the resolution of this domain to examine the role of species-specific positions, if a significant difference in primary sequence with E.coli is observed in this region.

We have included Supplemental Figure 8 with C1, C2 and C6 symmetry applied during reconstructions for the MlaD region. We did obtain a slightly improved resolution (3.7 Å Vs 3.9 Å), but the map did not include additional information that could help interpret the structure.

Other Remarks:

- Figure 1: from the figure on panel B it is not easy to identify MlaE. Maybe a coloring of the MlaE name with the same color as the 2 subunits would help better understand the architecture of the complex. Same comment for MlaF and MlaB. The orientation are well chosen and this figure will be otherwise informative once the subunits are properly identified

We tried to match colors to ref. 8 to enable direct comparison.

- Figure 4A: the Mg ion could be indicated and be displayed in another colour

We added a yellow sphere for the Mg atom.

- Figure 6: panel C is missing and the legend mentions the “transport” step whereas panel B mentions “transfer”.

We fixed both the wrong term and the missing panel

- Supplemental figure 3 and 4: panels A, B, C are not specified in the legend. In addition the particles are difficult to distinguish in image A. A selection of 2D classes would be more informative.

We have revised the figure legend as suggested.

- The authors refer alternatively to the 6-mer head of MalD as a crown, or basket, it would more consistent to stick to one name.

We changed “crown” to “basket”.

- page 7 line 176 (discussion): the authors mention “while this manuscript was in review”, maybe it would be more relevant to say “while this manuscript was in preparation” in the context of this submission ?

We submitted an initial version of this manuscript to a different journal, in May 2020 (see <https://doi.org/10.1101/2020.05.30.125013>). This statement therefore reflects the lengthy review process that this manuscript went through.

From the eLife reviews:

- I am not an MD simulations expert so I struggled to assess the significance of seeing events happen within a given number of nanoseconds (and this seemed awfully fast to me).

Molecular dynamics can faithfully produce lipid diffusion rates with the forcefield and methods we have used here, thus we are confident that the kinetics are realistic.

- The authors perform MD simulations on the MlaD basket presumably after replacing the DDM molecules with lipids. How this was done is not described at all.

We have now clarified this in the methods section.

Further, what happens if a simulation is run in the absence of lipid molecules at the MlaD protomer interfaces? Do bilayer lipids diffuse to these locations spontaneously?

One set of simulations were performed by removing the lipids from these locations and indeed spontaneous diffusion of lipids to these sites was observed, this is now explicitly stated in the manuscript.

REVIEWERS' COMMENTS:

Reviewer #2 (Remarks to the Author):

I am satisfied with the revisions that have been made.

Reviewer #3 (Remarks to the Author):

The authors of the article "Structure and lipid dynamics in the maintenance of lipid asymmetry (MLA) inner membrane complex of *A. baumannii*" submitted to Communications Biology have appropriately addressed the questions that were raised. In particular, figures have been improved and the main text has been clarified and corrected. A few minor points as listed below could still be addressed but that do not require major edits.

This study is complementary to other recent articles in the field and overall should be of interest for the readers of Communications Biology

Minor details:

- The supplemental figures have been abbreviated "Suppl. Fig." throughout the main text except for supplemental figure 7 (line 130)
- Unless I missed it, there is no sentence mentioning the new supplemental figure S9 in the main text. Maybe it should appear in the concluding remarks line 263 or in Figure 5 legend with a comment?